# Early Results of a Screening Program for Skin Cancer in Liver Transplant Recipients: A Cohort Study

**DOI:** 10.3390/cancers16061224

**Published:** 2024-03-20

**Authors:** Delal Akdag, Allan Rasmussen, Susanne Dam Nielsen, Dina Leth Møller, Katrine Togsverd-Bo, Emily Wenande, Merete Haedersdal, Hans-Christian Pommergaard

**Affiliations:** 1Department of Surgery and Transplantation, Centre for Cancer and Organ Diseases, Copenhagen University Hospital-Rigshospitalet, 2100 Copenhagen, Denmark; guenes.delal.akdag.01@regionh.dk (D.A.); allan.rasmussen@dadlnet.dk (A.R.);; 2Hepatic Malignancy Surgical Research Unit (HEPSURU), Department of Surgery and Transplantation, Centre for Cancer and Organ Diseases, Copenhagen University Hospital-Rigshospitalet, 2100 Copenhagen, Denmark; 3Viro-Immunology Research Unit, Department of Infectious Diseases, Copenhagen University Hospital-Rigshospitalet, 2100 Copenhagen, Denmark; dina.leth.moeller@regionh.dk; 4Department of Clinical Medicine, Faculty of Health and Medical Sciences, University of Copenhagen, 2200 Copenhagen, Denmark; mhaedersdal@dadlnet.dk; 5Department of Dermatology, Copenhagen University Hospital-Bispebjerg and Frederiksberg Hospital, 2400 Copenhagen, Denmark; katrinetogsverd@hotmail.com (K.T.-B.); emily.cathrine.wenande@regionh.dk (E.W.)

**Keywords:** liver transplantation, skin cancer, screening, incidence, risk factors

## Abstract

**Simple Summary:**

Skin cancer is the most common cancer in transplant recipients; however, screening may reduce advanced disease. The study aimed to determine referral rates to screening, the incidence, and risk factors of skin cancer in a Danish liver transplant recipient cohort. Of the 246 recipients, 89% were referred to screening and 15.6% were diagnosed with skin cancer or preneoplastic lesions during the screening period. Basal cell carcinoma was the most common skin cancer type followed by squamous cell carcinoma. Actinic keratosis was the most common preneoplastic lesion, followed by Bowen’s disease. The time since transplantation and actinic keratosis were identified as independent risk factors of skin cancer. The study determined the incidence and risk factors of skin cancer/preneoplastic lesions in liver transplant recipients enrolled in a screening program, while demonstrating a high screening referral rate.

**Abstract:**

(1) Background: Skin cancer is the most common cancer in transplant recipients. Timely and regular screening may reduce advanced disease. The study aimed to determine referral rates to screening, the incidence, and risk factors of skin cancer in a Danish liver transplant recipient cohort. (2) Methods: All first-time liver transplant recipients, >18 years old, attending outpatient care between January 2018 and December 2021 were included. The referral rates and incidence of skin cancer/preneoplastic lesions were calculated. Risk factors were assessed using Cox regression analyses. (3) Results: Of the 246 included recipients, 219 (89.0%) were referred to screening, and 102 skin cancer/preneoplastic lesions were diagnosed in 32 (15.6%) recipients. The IR of any skin cancer/preneoplastic lesion was 103.2 per 1000 person-years. BCC was the most frequent skin cancer followed by SCC, IR: 51.3 vs. 27.1 per 1000 person-years, respectively. No cases of MM were observed. The IR of actinic keratosis and Bowen’s Disease were 48.1 vs. 13.2 per 1000 person-years, respectively. Time since transplantation was independently associated with skin cancer/preneoplastic lesions, HR (95%CI) 2.81 (1.64–4.80). (4) Conclusions: The study determined the incidence and risk factors of skin cancer/preneoplastic lesions in liver transplant recipients enrolled in a screening program, while demonstrating a high screening referral rate.

## 1. Introduction

Skin cancer is the most common type of cancer reported in liver transplant recipients and the risk of skin cancer is more than 15 times higher than in the general population [1,2,3,4]. In addition, advanced and metastatic disease is more often observed in liver transplant recipients [2,4,5]. A key driving factor is considered to be the immunosuppressive treatment used to prevent organ rejection, which, in turn, reduces the immunological cancer surveillance [6,7]. Consequently, current clinical practice guidelines recommend annual skin cancer screening in all solid organ transplant (SOT) populations, including liver transplant recipients [8,9]. A recent multidisciplinary expert consensus guideline by dermatologists and transplant physicians furthermore highlights the importance of skin cancer risk stratification, a multidisciplinary approach, and patient education [10].

Adherence to regular dermatological assessment is reported to reduce the rates of advanced skin cancer in SOT recipients [11,12]. However, studies reporting on the incidence and risk factors of skin cancer in liver transplant recipients enrolled in a skin cancer screening program are sparse [13,14,15,16,17,18,19,20,21]. Due to significant variations in skin cancer risk among different SOT groups, it is essential to investigate the risk within distinct subpopulations [22,23].

The primary aim of the study was to determine referral rates to dermatological skin cancer screening and incidence of skin cancer/preneoplastic lesions in a Danish liver transplant cohort. Secondly, we aimed to identify risk factors associated with skin cancer in liver transplant recipients.

## 2. Materials and Methods

The study was reported according to The Strengthening the Reporting of Observational Studies in Epidemiology (STROBE) statement [24].

### 2.1. Study Design and Setting

The study was a cohort study, established on the 1 January 2018, and inclusion is ongoing. We conducted an evaluation of the cohort study between 1 January 2018 and 31 December 2021.

Since 1 January 2018, liver transplant recipients attending outpatient care at the Department of Surgery and Transplantation, Rigshospitalet in Copenhagen, Denmark, regardless of the time since transplantation, have been routinely referred to dermatological skin cancer screening. The referred recipients attended screening at either a dermatological department in a hospital setting or a private dermatological practice. During screening visits, recipients underwent a full-body skin exam, including the assessment of genital skin by an experienced clinician. Dermatoscopy of suspicious lesions was routinely performed at visits. Screening intervals depended on recipient risk profiles (i.e., history of skin cancer, presence of clinical actinic keratosis/photodamage, Fitzpatrick skin type, and age). Generally, high-risk patients were seen every 3–6 months, intermediate-risk patients annually, and low-risk patients every 12–24 months. Visits were expedited upon discovery of any new lesions by patients. Prior to this era, referral to skin cancer screening was performed on a case-by-case basis depending on clinical indication.

### 2.2. Study Participants—Inclusion and Exclusion Criteria

All first-time liver transplant recipients, >18 years of age at the time of transplantation, attending outpatient care at the Department of Surgery and Transplantation Rigshospitalet, were included in the study. The remaining Danish liver transplant recipients residing in Jutland and Funen were followed locally and not included in the current study. Retransplanted- and multi-organ recipients were excluded from the study, due to increased use of immunosuppressive treatment. Also excluded were liver transplant recipients with <1 year of follow-up after transplantation, since the first dermatological skin cancer screening typically occurred after this period.

### 2.3. Data Sources

Data on skin cancer (here defined as basal cell carcinoma (BCC), squamous cell carcinoma (SCC) and malignant melanoma (MM)) and preneoplastic lesions (here defined as actinic keratosis (AK) and Bowen’s disease (BD) (squamous cell carcinoma in situ)) were based on histopathological data from the national Danish Pathology Data Bank (Patobank) [25]. Patobank is a nationwide, real-time database with data transfer from all Danish departments of pathology. Pathology data are linked to a national civil registration number, assigned to all Danish residents, and controlled by the Danish Data Protection Agency.

Risk factor data, including information on clinical sun exposure (defined as reported outdoor work, leisure exposure and clinical signs of sun damage/actinic degeneration) and UVR sensitivity in terms of Fitzpatrick skin type, were determined and recorded by a dermatologist during skin cancer screening [26]. Other demographic data, including indication for liver transplantation, smoking status and referral rates were extracted from the Knowledge Center for Transplantation database, a national database on clinical characteristics in solid organ transplant recipients [27].

### 2.4. Outcome Measures

The study’s outcome measures were referral rates to dermatological skin cancer screening, incidence proportion and incidence rate of skin cancers and preneoplastic lesions. The referral rate was determined as the proportion of liver transplant recipients attending skin cancer screening. The incidence proportions of skin cancer/preneoplastic lesions were determined as proportions of new cases in the study population during the screening period. The incidence rates of skin cancer/preneoplastic lesions were determined as new cases per 1000 person-years. Each recipient was followed from the date of the first skin cancer screening until either the date of the first skin cancer or the end of the study period on 31 December 2021, whichever came first. Any histologically confirmed skin cancer/preneoplastic lesion, along with number of lesions and date of diagnosis, was recorded for all liver transplant recipients enrolled for screening. To determine the incidence, only skin cancer/preneoplastic lesions debuting after the date of the first skin cancer screening were registered. To avoid double counting, skin cancers diagnosed by punch biopsy and subsequently excised were counted as one tumor, determined by the proximity of analysis dates (<3 months) and identical anatomical localization. Also excluded were recurrent lesions, as designated by pathology codes.

The secondary outcome of the study was to determine the potential risk factors of skin cancer/preneoplastic lesions. Independent variables included in the model with skin cancer/preneoplastic lesions combined and preneoplastic lesions alone as the outcomes were age, time since transplantation, and dermatological assessment <2 years after transplantation. The model with skin cancer alone included independent variables were age, time since transplantation, and actinic keratosis.

In the model assessing the risk factors of skin cancer alone, we initially included dermatological assessment <2 years after transplantation as a risk factor. However, the model exhibited separation, wherein dermatological assessment <2 years after transplantation perfectly predicted the absence of skin cancer (e.g., all cancer cases received dermatological assessment more than 2 years after transplantation), resulting in infinite parameter estimates. To address this issue and ensure model stability, we chose to exclude the variable from the final analysis.

### 2.5. Statistical Analysis

Non-normally distributed continuous data were reported as medians with interquartile ranges (IQRs). Categorical data were reported as frequency counts and percentages (%) of subjects within each category. *p*-values were calculated using Chi squared or Fisher’s exact test. *p*-values were considered statistically significant if *p* ≤ 0.05. To identify the potential risk factors associated with skin cancer/preneoplastic lesions, hazard ratios (HRs) and 95% confidence intervals (95% CIs) were computed using the Cox proportional hazards model. The variables with a statistically significant association upon univariable analysis were included in a multivariable model. All analyses were performed using RStudio 2022.07.1+554.

## 3. Results

### 3.1. Evaluation of the Screening Program during the Screening Period

#### 3.1.1. Referral Rates

At the study initiation, 246 liver transplant recipients were included in the study. Of these, 66.7% resided in Capital Region and 33.3% in Region Zealand (Figure 1).

In total, 219 (89.0%) liver transplant recipients were referred to dermatological skin cancer screening. Among them, 205 (93.6%) recipients were followed at a dermatological department in a hospital setting, while 14 (6.3%) were followed in a private dermatological practice. Recipients followed at a dermatological department in a hospital setting (n = 205) had available data on potential risk factors and were therefore included for further analysis.

#### 3.1.2. Visiting Frequency during the Screening Period

When considering the recipients followed at a dermatological department in a hospital setting (n = 205), the median (IQR) number of visits was 2.0 (1.0–3.0) and the proportion of recipients with 2 or more visits was 61.9%. Figure 2a illustrates the range of visiting frequencies.

#### 3.1.3. Years from Liver Transplantation to the First Screening

The time from liver transplantation to the first dermatological skin cancer screening is presented in Figure 2b. The screening program was implemented for all recipients seen in the outpatient clinic, regardless of the time since transplantation. Thus, among the recipients included in the screened population, a considerable range in time since transplantation was noted. The median (IQR) number of years from liver transplantation to the first visit skin cancer screening was 5.0 (1.0–10.0); however, in recipients transplanted after the implementation of the screening program (2018), the median (IQR) number of years was 1.0 (0.0–1.0). Overall, less than half of the recipients had their first skin cancer screening within 2 years after liver transplantation (38.3%). In contrast, in recipients transplanted after implementation of the screening program, this proportion was 98.0%.

### 3.2. Patient Characteristics

#### Patient Characteristics

Of 205 liver transplant recipients followed at a dermatological department in a hospital setting, 32 (15.6%) developed skin cancer/preneoplastic lesion, with 102 skin cancer/preneoplastic lesions in total diagnosed over a median (IQR) of 2 (0.0–3.0) years of follow up (Table 1). Among the 32 recipients, the average number of skin cancer/preneoplastic lesions per patient was 3.2, while the highest number of observed cancers diagnosed in a single recipient was 10. A detailed characterization of liver transplant recipients with skin cancer/preneoplastic lesion is presented in Appendix A.

Recipients with skin cancer/preneoplastic lesions were significantly older than the non-cancer group (median age 67.4 vs. 55.7 years, *p* < 0.01) and had been transplanted for a longer time (median 14 vs. 6 years, *p* < 0.01). Correspondingly, the skin cancer/preneoplastic lesion group had a longer follow-up time in the screening program compared to the non-cancer group (median 3 vs. 1 years, *p* < 0.01) and more often had ≥ 2 dermatological visits compared to the non-cancer group (88.9% vs. 57.6%, *p* < 0.01). There was no significant difference in sun exposure, smoking status, current immunosuppression, and Fitzpatrick skin type between the two groups (Table 1).

### 3.3. Skin Cancer Incidence and Risk Factors

#### 3.3.1. Incidence Proportions and Rates

During the screening period, with a median (IQR) follow-up of 3 years (1.0–3.0), a total of 205 liver transplant recipients contributed with a total of 335.9 person-years. None were lost to follow-up (Table 2). Of all recipients, 24 developed skin cancer, giving an incidence proportion (IP) of 11.7%. The IP of skin cancer/preneoplastic lesions combined was 15.6% (32/205). BCC was the most common skin cancer, with an IP of 8.8%, followed by SCC with an IP of 4.9%. None of the recipients developed MM during the screening period, with an IP of 0.0%. Of the preneoplastic lesions, AK was the most common preneoplastic lesion, with an IP of 8.3%, whereas the IP of BD was 2.4%.

The incidence rate (IR) of skin cancer was 71.5 per 1000 person-years, and for skin cancer preneoplastic lesions, the combined IR was 103.2 per 1000 person-years. The IRs of BCC and SCC were 51.3 per 1000 person-years and 27.1 per 1000 person-years, respectively. For preneoplastic lesions, the IRs of AK and BD were 48.3 per 1000 person-years and 13.2 per 1000 person-years, respectively.

#### 3.3.2. Risk Factors for Skin Cancer/Preneoplastic Lesions

Uni- and multivariable analyses with skin cancer alone as a dependent variable are presented in Table 3. In univariable analysis, age per decade increase, time since transplantation per decade increase, and actinic keratosis were associated with an increased risk of skin cancer (HR (95% CI) 2.41 (1.56–3.71), HR (95% CI) 3.15 (2.01–4.94), and HR (95% CI) 8.81 (3.88–20.00), respectively). However, these did not remain significant in multivariable analyses.

Uni- and multivariable analyses with preneoplastic lesions alone as a dependent variable are presented in Table 4. In multivariable analyses, time since transplantation per decade increase was associated with an increased risk of preneoplastic lesions (HR (95% CI) 3.19 (1.65–6.18)). In univariable analysis, age per decade increase was associated with an increased risk of preneoplastic lesions (HR (95% CI) 2.72 (1.65–4.48)), and dermatological skin cancer screening < 2 years after transplantation was associated with a decreased risk of preneoplastic lesions (HR (95%CI) 0.15 (0.03–0.63)). However, these did not remain significant in multivariable analyses.

Uni- and multivariable analyses with skin cancer/preneoplastic lesions combined as a dependent variable are provided in Table 5. In multivariable analyses, time since transplantation per decade increase was associated with an increased risk of skin cancer/preneoplastic lesions (HR (95% CI) 2.81 (1.64–4.80)). In univariable analyses, age per decade increase was associated with an increased risk of skin cancer/preneoplastic lesions (HR (95%CI) 2.52 (1.71–3.72)), and dermatological skin cancer screening < 2 years after transplantation was associated with a decreased risk of skin cancer/preneoplastic (HR (95%CI) 0.09 (0.02–0.37)); however, this finding did not remain significant in the multivariable analysis.

## 4. Discussion

In this study, we determined the referral rates for skin cancer screening and incidence of skin cancer/preneoplastic lesions in liver transplant recipients four years after the initiation of a screening program. During the study period, we found that nearly 90% of the recipients were referred for dermatological skin cancer screening. A total of 102 skin cancers/preneoplastic lesions were found in 32 patients (15.6%). Time since transplantation per decade increase was associated with an increased risk of skin cancer/preneoplastic lesions and preneoplastic lesions alone in multivariable analysis. Dermatological skin cancer screening < 2 years after transplantation was significantly associated with a decreased risk of skin cancer/preneoplastic lesions in univariate analysis. A possible explanation for this could be that recipients who were transplanted more recently, and thus closer to the introduction of the screening program, were more likely to undergo early skin cancer screening and that recently transplanted recipients were younger. However, this did not remain significant in multivariable analysis.

Relatively sparse literature exists on the incidence rate of skin cancer in liver transplant recipients specifically. The existing studies are mostly retrospective and only few involved actual screening programs [11,12,16,17,18,28]. The herein reported incidence rate falls within the range of prior reports on skin cancer/preneoplastic lesions in liver transplant recipients [21,29,30]. Herrero et al. observed an incidence of 43.3 skin cancer cases per 1000 person-years among 170 liver transplant recipients enrolled in a skin cancer screening program residing in Spain or Portugal [21]. All recipients were screened by a dermatologist at 6 and 12 months after transplantation and every year thereafter. The incidence rate reported in the present study is higher than that reported by Herrero et al. This may be attributed to the exclusion of 52 liver transplant recipients out of 222 eligible individuals based on factors such as survival less than 6 months and refusal to participate in skin cancer screening. By excluding the frailest recipients and possibly those with the highest disease burden, selection bias may have been introduced. Additionally, the notable differences in skin phototype and consequent skin cancer risk exist between southern and northern European populations. Esfeh et al. conducted a retrospective cohort analysis of 998 liver transplant recipients in Cleveland, United States [30]. The study reported an incidence rate of 84.8 skin cancer cases per 1000 person-years, aligning with the findings of the present study. However, based on tumor subtypes, authors report a lower BCC (33.2 per 1000 person-years) and SCC (36.9 per 1000 person-years), and a higher MM (14.8 per 1000 person-years) ratio. The retrospective design that relied on medical records might have led to underreporting of cancer cases. The demographic differences in our study populations may further explain these discrepancies.

Approximately 90% of liver transplant recipients included in the study were referred to skin cancer screening. A comprehensive study of 10,183 SOT recipients in Ontario, Canada (23.7% liver transplant recipients), assessed adherence to screening using physician consultation claims [11]. Higher adherence correlated with reduced advanced skin cancer defined as requiring surgical intervention. Furthermore, adherence was associated with a history of skin cancer/preneoplastic lesions, white race, and younger age at transplantation. Despite this, the overall adherence was low, which could reflect insufficient referrals to dermatological screening or patient behavior. In contrast to our study which specifically outlines the referral rates for liver transplant recipients, the study did not determine the rate of referrals.

Interestingly, the study observed a higher incidence ratio of BCC compared to SCC. This observation is contrary to previous SOT studies that typically report a higher SCC versus BCC incidences [4]. However, past studies primarily included kidney, heart, and lung transplant recipients, with comparatively fewer liver transplant recipients. Skin cancer trends differ among liver transplant recipients compared to other SOT recipients [22,23]. Prior studies specifically addressing skin cancer incidence in liver transplant recipients support the herein reported BCC:SCC ratio [20,21,30]. Further investigation is needed to determine if this pattern is specific to liver transplant recipients or influenced by the high screening attendance in our cohort, possibly indicating a protective effect through the regular detection and treatment of preneoplastic lesions and patient education.

This investigation provides insights into (1) skin cancer incidence observed four years after program initiation and (2) the associated risk factors for skin cancer/preneoplastic lesions specifically in liver transplant recipients. Furthermore, the study determined the degree of the program’s implementation based on the referral rates and number of screening visits. Long-term screening data are of great interest to dermatologists and transplant physicians [8,9,10]. We found that the burden of disease varied considerably between patients (one patient had 10 lesions), underscoring the need for risk stratification and individualized screening of high- and low-risk recipients, as prescribed by current recommendations [10].

The primary study limitations included a short follow-up of only a median of three years and a small sample size. Despite these limitations, the study successfully identified the time since transplantation and actinic keratosis as significant risk factors for skin cancer, as reported previously [15,31,32,33]. On the other hand, the study’s failure to identify and validate other known risk factors including male sex, fair skin, age, and indication for liver transplantation, may be due to the small sample size [11,16,30]. The study primarily included liver transplant recipients of Northern European ethnicity with a predominance of fairer skin phototypes and thereby a higher skin cancer risk, which might affect the generalizability of the study. An additional limitation concerns our calculated incidence rates for actinic keratosis. The diagnosis of this lesion is typically based on clinical evaluation alone. As such, by including only histologically verified diagnoses, the study likely underestimates the true incidence rate of actinic keratosis in the cohort.

## 5. Conclusions

In conclusion, this screening study contributed with data on the incidence and risk factors of skin cancer/preneoplastic lesions in liver transplant recipients enrolled in a screening program—a sparsely covered area in the literature—while demonstrating a high screening referral rate. Considering the short study follow-up, additional long-term data on our expanding cohort are needed for improved insight into the effects of the herein-described screening program.

## Figures and Tables

**Figure 1 cancers-16-01224-f001:**
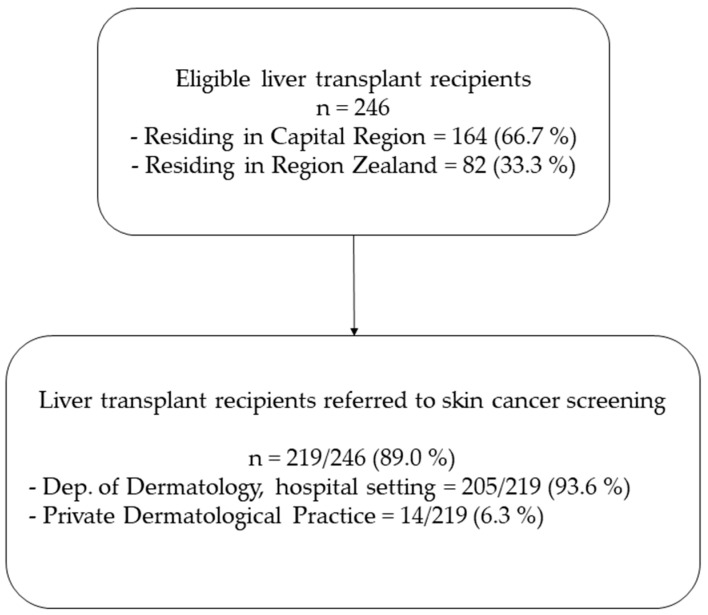
Liver transplant recipients and referral to skin cancer screening. Liver transplant recipients followed at a dermatological department in a hospital setting were included for further analyses (n = 205).

**Figure 2 cancers-16-01224-f002:**
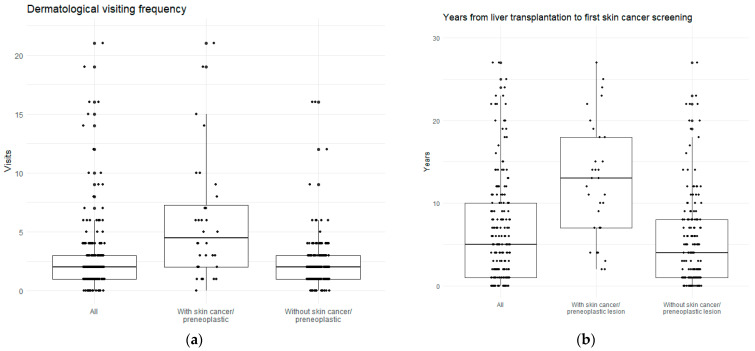
Data includes all liver transplant recipients followed at a department of dermatology in a hospital setting. (**a**) Dermatological visiting frequencies for all recipients and, recipients with or without skin cancer/preneoplastic lesions. (**b**) Years from liver transplantation to first skin cancer screening for all recipients and, recipients with or without skin cancer/preneoplastic lesions.

**Table 1 cancers-16-01224-t001:** Patient characteristics.

Patient Characteristic	Recipients without Skin Cancer/Preneoplastic Lesions, n = 173	Recipients with Skin Cancer/Preneoplastic Lesions, n = 32	*p*
Age, median (IQR)	55.7 (47.2–63.5)	67.4 (60.8–73.6)	<0.01 *
Sex, male, n (%)	98 (56.6%)	20 (62.5%)	0.67
Age at LTX•, yrs., median (IQR)	48.9 (39.0–55.0)	49.6 (42.9–55.7)	0.45
Time since LTX, yrs., median (IQR)	6.0 (3.0–10.0)	14.0 (11.8–23.3)	<0.01 *
Follow-up, yrs., median (IQR)	1.0 (0.0–3.0)	3.0 (2.0–3.0)	<0.01 *
Number of visits to dermatologist, median, (IQR)	2.0 (1.0–3.0)	5.0 (3.0–7.5)	<0.01 *
Dermatological visits, n (%) ≥2 visits <2 visits	98/170 (57.6%) 72/170 (42.4%)	24/27 (88.9%) 3/27 (9.4%)	<0.01 *
Dermatological skin cancer screening ≤ 2 yrs. after LTX, n (%)	70/157 (44.6%)	2/31 (6.5%)	<0.01 *
Indication for LTX, n (%) Cirrhosis (alcoholic and cryptogenic) Primary sclerosing cholangitis Primary biliary cholangitis Hepatitis C Hepatocellular carcinoma Other	40/173 (23.1%) 50/173 (28.9%) 13/173 (7.5%) 3/173 (1.7%) 13/173 (8.0%) 45/173 (26.0%)	5/32 (15.6%) 5/32 (15.6%) 3/32 (9.4%) 0/32 (0.0%) 1/32 (3.1%) 15/32 (46.9%)	0.96
Smoking status, n (%) Current Former Never	22/171 (12.9%) 64/171 (37.4%) 85/171 (49.7%)	4/31 (22.6%) 7/31 (22.6%) 16/31 (51.6%)	0.19
Current immunosuppression, n (%) CNI (Tacrolimus, Ciclosporin) mTOR inhibitor (Everolimus) Mycophenolate Azathioprin Corticosteroid	157/173 (90.8%) 17/173 (9.8%) 131/173 (75.7%) 13/173 (7.5%) 81/173 (46.8%)	29/32 (90.6%) 2/32 (6.3%) 23/32 (71.9%) 5/32 (15.6%) 16/32 (50.0%)	0.66
Sun exposure, n (%) † Outdoor work Leisure exposure Sun damage/Actinic degeneration	17/69 (24.6%) 29/72 (40.3%) 26/80 (32.5%)	4/12 (33.3%) 9/14 (64.3%) 10/14 (71.4%)	0.75
Fitzpatrick skin phototype, n (%) I, II, III IV, V, VI	57/68 (83.8%) 11/68 (16.2%)	9/11 (81.8%) 2/11 (18.2%)	1.00

* Significant *p*-values. • LTX: liver transplantation. † Outdoor work incl. yes, previous, and partly. Leisure exposure includes persons categorized as outdoor person and regular sunbathers. Sun damage/actinic degeneration incl. moderate and severer categories.

**Table 2 cancers-16-01224-t002:** Incidence of skin cancer/preneoplastic lesions.

Incidence	Incidence Proportion, (%)	Incidence Rate, per 1000 Person-Year
Skin cancer (BCC, SCC, or MM) BCC SCC MM	24/205 (11.7%) 18/205 (8.8%) 10/205 (4.9%) 0/205 (0.0%)	71.5 51.3 27.1 0.0
Preneoplastic lesions (actinic keratosis or Bowen’s disease) Actinic keratosis Bowen’s Disease	20/205 (9.6%) 17/205 (8.3%) 5/205 (2.4%)	58.0 48.3 13.2
Skin cancer/preneoplastic lesions (incl. BCC, SCC, MM, actinic keratosis, or Bowen’s disease)	32/205 (15.6%)	103.2

**Table 3 cancers-16-01224-t003:** Risk factor analyses for skin cancer.

Risk Factors for Skin Cancer	Univariable HR (95% CI)	*p*	Multivariable HR (95% CI)	*p*
Age, pr. decade	2.41 (1.56–3.71)	<0.01 *	1.44 (0.89–2.32)	0.14
Time since LTX, pr. decade	3.15 (2.01–4.94)	<0.01 *	1.61 (0.81–3.21)	0.18
Actinic keratosis, yes vs. no	8.81 (3.88–20.00)	<0.01 *	2.75 (0.81–9.40)	0.11

* Significant *p*-values.

**Table 4 cancers-16-01224-t004:** Risk factor analyses for preneoplastic lesions.

Risk Factors for Preneoplastic Lesions	Univariable HR (95% CI)	*p*	Multivariable HR (95% CI)	*p*
Age, pr. decade	2.72 (1.65–4.48)	<0.01 *	1.52 (0.91–2.55)	0.11
Time since LTX, pr. decade	4.31 (2.64–7.02)	<0.01 *	3.19 (1.65–6.18)	<0.01 *
Dermatological skin cancer screening ≤ 2 yrs. after LTX, yes vs. no	0.15 (0.03–0.63)	0.01 *	0.93 (0.16–5.44)	0.94

* Significant *p*-values.

**Table 5 cancers-16-01224-t005:** Risk factor analyses for skin cancer/preneoplastic lesions.

Risk Factors for Skin Cancer/Preneoplastic Lesions	Univariable HR (95% CI)	*p*	Multivariable HR (95% CI)	*p*
Age, pr. decade	2.52 (1.71–3.72)	<0.01 *	1.46 (1.00–2.18)	0.06
Time since LTX, pr. decade	4.18 (2.80–6.25)	<0.01 *	2.81 (1.64–4.80)	<0.01 *
Dermatological skin cancer screening ≤ 2 yrs. after LTX, yes vs. no	0.09 (0.02–0.37)	<0.01 *	0.36 (0.08–1.79)	0.21

* Significant *p*-values.

## Data Availability

The data presented in this study are available in this article (and Appendix A).

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
