# Peer review of "Early Results of a Screening Program for Skin Cancer in Liver Transplant Recipients: A Cohort Study"

_cancers, 2024, doi:10.3390/cancers16061224_

Round 1
Reviewer 1 Report
Comments and Suggestions for Authors
very good and well reported study which will provide important insights and reference for clinical practice. Actually, I have no special comments.
Only a minor remark, I think that the Authors can expand on the relevance of their findings in countries other than Nothern Europe.
well done!
Comments on the Quality of English LanguageFine, only minor polishing at the proof-reading stage
Reviewer 2 Report
Comments and Suggestions for Authors
The authors presented an interesting article, describing the referral rates to dermatologic screening and incidence of skin cancer among liver transplant recipients. Despite significant findings, which are very relevant to recent research on the field, in my view, there are important issues needed to be addressed by the authors. The main areas need amendments are methods and results.
Methods:
-inclusion and exclusion criteria: clearly defined. Please amend in Fig. 1 that only 205 were included in the analysis (14 patients were excluded due to missing values?). Also, was data about previous immunosuppression available and if affected the outcomes?
-How screening was conducted and how often? Please state if the screening was annually, or two times per year, how the patients were followed up (i.e. simple dermatologic examination, with dermatoscope or with total body photography?). Also, if there was a more intensive follow up for patients with skin cancer (which could possibly explain the higher number of visits in those patients) or if the follow up program was different between patients with and without skin cancer.
Similarly, s190- 192, why patients with skin cancer had higher median f.u after screening compared to non-skin cancer patients?
-Another point is why age was excluded from the analysis (s130-132). If it was highly correlated with time since transplantation (but first the authors might prove that providing correlation plot), it would be excluded immediately from multivariate models. Also, age referred to age at first or last f.u or age at skin cancer diagnosis please state. Also, it is weird why age is statistically different, but age at transplantation is similar between groups. Please explain.
-s162-163. Number of visits >2 according to s162-163 and table 1 did not sum up in patients (61.5% of patients had >/= 2 visits, but that number did not sum up with patients referred to table 1). Also, it would be useful to provide boxplots (instead of lines in fig2a) for both visits and years since transplantation for the overall cohort and 2 groups (with skin and without skin cancer) in order to render differences and outliers more evident.
-Risk factors: in model including only skin cancer patients, why dermatologic visits and dermatologic screening <2 years were not included. Also, are there any other variables included in univariate analysis (a supplementary table with all variables included in univariate analysis would be supporting). Also, please use 2 decimals for OR and 95%CI. In addition to, was a separate analysis for BCC or SCC conducted? Also, if presence of actinic keratosis influenced the risk of SCC alone or both BCC and SCC?
Another point is the analysis for premalignant lesions only. Was that analysis conducted?
Comments on the Quality of English LanguageMinor editing needed
Reviewer 3 Report
Comments and Suggestions for Authors
I have appreciated your work on this interesting topic. Liver transplant is a reality and also it's prowess to developing skin cancers. Your study and statistical analysis is well made. The findings regarding BCC to SCC ratio are surprising. Your article comes to shed a little light in a not enough covered aspect of evolution under transplant.
Round 2
Reviewer 2 Report
Comments and Suggestions for Authors
The authors amended adequately most of the comments.
No more comments. Well done!